# $\epsilon$-Best-Arm Identification in Pay-Per-Reward Multi-Armed Bandits

**Sivan Sabato**
Department of Computer Science
Ben-Gurion University of the Negev
Beer-Sheva, Israel 8410501
sabatos@cs.bgu.ac.il

## Abstract

We study $\epsilon$-best-arm identification, in a setting where during the exploration phase, the cost of each arm pull is proportional to the expected future reward of that arm. We term this setting *Pay-Per-Reward*. We provide an algorithm for this setting, that with a high probability returns an $\epsilon$-best arm, while incurring a cost that depends only linearly on the total expected reward of all arms, and does not depend at all on the number of arms. Under mild assumptions, the algorithm can be applied also to problems with infinitely many arms.

## 1 Introduction

Consider placing an ad next to search engine results, based on the search query. In a preliminary survey for a future promotion, a retailer wishes to identify the best query expression to link to its ad, that is, the expression that maximizes the expected number of clicks on the ad. The payment per ad is proportional to the number of clicks. However, during the survey, clicks do not lead to profit, while the payment is still due.

This problem can be formulated as a type of best-arm-identification problem in a stochastic multi-armed bandit (MAB) setting [see, e.g., 14, 15, 8, 3]. In stochastic MAB, there is a set of arms, each associated with a non-negative reward distribution. Each pull of an arm draws an instantaneous reward from the corresponding reward distribution. The MAB algorithm iteratively pulls arms and observes their instantaneous rewards. At the end of the run, the algorithm selects one arm. The goal is to find an arm with a near-maximal expected reward with a high probability. In the example above, each arm in the MAB framework can be mapped to a specific search expression, and pulling the arm is equivalent to linking the ad to the expression represented by this arm for a given period of time, and then observing the number of clicks. In standard MAB, each arm pull incurs a unit cost. However, in the example above, the cost of pulling an arm is the number of clicks on the ad, assuming the common pay-per-click advertisement model. Thus, the expected cost in the survey stage is proportional to the expected reward after the survey stage. We term this setting *Pay-Per-Reward* (PPR). Other applications of this model include any case where there is an exploration stage in which rewards are not collected, but payment is still proportional to the quality of the arm. For instance, consider a sensor field, where the goal is to find the most active sensor, but each activity report has a communication cost. Here, "pulling an arm" is done by activating the sensor's reporting option for a given period of time. Another example is finding the best crowd worker on a crowd-sourcing platform. Here, an "arm pull" is equivalent to providing the worker with a test task. The workers are paid according to their success rate in the test task, thus a more successful worker will be paid more during the exploration stage.

*Our contribution.* We provide an algorithm, MAB-PPR, for the pay-per-reward setting, that finds an $\epsilon$-optimal arm with a probability of at least $1 - \delta$, for given $\epsilon, \delta \in (0, 1)$. We show that the cost

incurred by MAB-PPR has no dependence on the number of arms, and only a linear dependence on the total expected reward of all arms. Our results generalize beyond bounded reward distributions, and support some heavy-tailed distributions with bounded second moments. Under mild assumptions, the algorithm can be applied also to problems with infinitely many arms.

Since the cost of MAB-PPR is independent of the number of arms, it is especially useful when the number of arms is potentially very large, while the total expected reward is bounded. For instance, in the ad-placement example, the number of arms is proportional to the number of admissible search expressions, which can be practically unbounded, but only a small number of those are expected to have large click rates when linked to a specific ad. Other applications in which the number of arms can be very large include, for instance, recommender systems [32], content personalization [24] and web-content optimization [1].

For the standard setting, in which each arm pull has a unit cost, [14] provided an $\epsilon$-best-arm identification algorithm with a sample complexity that depends linearly on the number of arms, using an algorithm that they term the median-elimination algorithm. This algorithm halves the number of candidate arms in each round based on their empirical mean rewards. Instead, in our setting, one should try to halve the total expected reward of the arms that remain in each round. This requires a new estimation scheme. Another challenge in the PPR setting is that due to the potentially unbounded number of arms, sums of instantaneous rewards of sets of arms do not have a bounded support, even if single-arm rewards are bounded. Moreover, in some settings one might prefer to simultaneously pull an entire set of arms from the joint distribution of rewards of all arms. Thus, standard assumptions on independence between arms do not necessarily hold.

MAB-PPR addresses these challenges using a two-step estimation procedure in each round, combined with heavy-tailed mean estimation. This guarantees success even under some dependence between arms. Support for some unbounded reward distributions of single arms is easily obtained within this framework, by using heavy-tailed mean estimation for the expected rewards of single arms as well. Our analysis shows that the PPR setting can be successfully handled in a wide range of regimes.

**Paper structure**   After discussing related work, we present the setting and preliminaries in Section 2. The main result is stated in Section 3. The MAB-PPR algorithm is listed and described in Section 4. We prove the main result in Section 5, and conclude in Section 6.

## 1.1   Related work

[14] formulated the stochastic MAB $\epsilon$-best-arm identification problem (also termed the PAC setting) for bounded rewards. They showed that the number of pulls can be linear in the number of arms, without logarithmic factors that would be necessary in a naive algorithm that pulls all arms the same number of times. Closely matching lower bounds for this setting were given in [22]. Instance-specific bounds in this setting were studied in [19, 17, 12]. A fixed-budget variant was studied in [8, 3, 10].

The works above assume the standard cost model for MAB, in which each arm pull costs one unit. A different cost model has been studied in the setting of budgeted MAB [25, 26, 13, 27, 29, 20]. Here, arms are associated with a random or deterministic cost, in addition to a reward distribution, and the goal is to maximize the cumulative reward within total cost constraints. [30] studied a best-arm variant of this problem. MAB with infinitely many arms, also termed many-armed bandits, has been studied in the unit-cost model both for the cumulative regret setting [5, 6, 28, 20] and for exploration settings [11, 4]. [20] study infinitely-many arms in the budgeted MAB setting.

A useful tool for mean estimation for heavy-tailed random variables is median-of-means estimation [2]. This tool has been recently used in various contexts in machine learning [e.g., 23, 7, 16, 21, 18]. MAB with heavy-tailed rewards under the standard unit-cost model was studied in the regret setting [9] and for pure exploration with unbounded rewards [31].

## 2   Setting and Preliminaries

For an integer $i$, denote $[i] := \{1, \ldots, i\}$. Let $K$ be the total number of available arms, and denote the set of available arms by $[K]$.[1] Let $X_1, \ldots, X_K$ be the random variables associated with the reward

distributions of the corresponding arms. We assume that the random variables are non-negative, and have a finite expectation and variance. Pulling arm $i$ is equivalent to drawing an independent copy of $X_i$ and observing its value. We study here a more general setting, in which sets of arms can be pulled simultaneously. We assume some joint distribution of the random vector $\mathbf{X} = (X_1, \ldots, X_K)$. A pull of a specific set of arms is equivalent to drawing an independent copy of $\mathbf{X}$ and observing the rewards of all the arms in the set. The standard setting, in which arms are pulled one by one, is equivalent to assuming that $X_1, \ldots, X_K$ are statistically independent in $\mathbf{X}$. Thus, we generally assume below that sets are simultaneously pulled.

Denote the expected reward of arm $i$ by $w_i := \mathbb{E}[X_i]$, termed below the arm's *weight*. Let $W := \sum_{i \in [K]} w_i$. For a set $A \subseteq [K]$, denote $X_A := \sum_{i \in A} X_i$ and $W_A := \sum_{i \in A} w_i$. Consider a run of the algorithm in which the total number of times each arm is pulled (regardless of whether some of the arms were pulled simultaneously) is $n$, and let $I_1, \ldots, I_n$ be the sequence of pulled arms. The cost of the algorithm in this run is $\sum_{i \in [n]} w_{I_i}$. We make two assumptions on the distribution of $\mathbf{X}$.

**Assumption 2.1.** *For all $i \in [K]$, $\sigma_i^2 := \mathrm{Var}(X_i) \leq w_i \leq 1$.*

This assumption clearly holds if $X_i$ are bounded in $[0, 1]$. Nonetheless, it is more general, and can hold also for reward distributions with unbounded support.

**Assumption 2.2.** *There is a constant $V \geq 0$, such that for any subset of the arms $A \subseteq [K]$, $\sigma_A^2 := \mathrm{Var}(X_A) \leq V W_A$.*

Assumption 2.2 trivially holds with $V = 1$ if arm-pulls of individual arms are statistically independent, as when they are pulled separately. In this case, $\sigma_A^2 = \sum_{i \in A} \sigma_i^2 \leq \sum_{i \in A} w_i = W_A$, where Assumption 2.1 was used to bound $\sigma_i^2$. If arm rewards are not statistically independent, then the existence and value of the constant $V$ depend on the distribution of $\mathbf{X}$. For instance, if each arm is positively correlated with at most $Q$ other arms, then $V$ can be upper bounded by $1 + 2Q$ (see the supplementary material, Appendix A). We note that our results below are indifferent to values of $V$ as large as $\frac{1}{\epsilon}$.

We use the *median-of-means* method [2] to estimate the mean of a distribution with a bounded variance. In this method, $q$ batches of $\ell$ independent samples are used; A separate empirical mean is calculated from each batch, and the estimate is the median of these means. Formally, the $(q, \ell)$-*MoM estimator* of a random variable $X$ from a sample of i.i.d. draws $\{x_{i,j}\}_{i \in [q], j \in [\ell]}$ of $X$, is defined as $\mathrm{Median}\{\frac{1}{\ell} \sum_{j \in [\ell]} x_{i,j} \mid i \in [q]\}$. We use the following formulation of the result of [2] on the convergence of the MoM estimator to the true mean, similarly to [16]. The proof is provided for completeness in the supplementary material, Appendix B.

**Lemma 2.3** (Median-of-Means). *Let $X$ be a random variable with mean $w$ and variance $\sigma^2 < \infty$. Let $q, \ell$ be integers, and let $\hat{w}$ be the $(q, \ell)$-MoM estimate of $w$ from a sample of $q\ell$ i.i.d. draws of copies of $X$. With a probability at least $1 - \exp(-\frac{2}{9}q)$, we have $|w - \hat{w}| \leq \sqrt{6\sigma^2/\ell}$.*

## 3 Main result

We propose the MAB-PPR algorithm, which accepts parameters $\epsilon, \delta \in (0, 1)$, iteratively pulls arms, and outputs a single arm. MAB-PPR satisfies the following guarantee.

**Theorem 3.1.** *Suppose that Assumptions 2.1 and 2.2 hold. Let $W := \sum_{i \in [K]} w_i$, and let $V$ as defined in Assumption 2.2. With a probability of at least $1 - \delta$, MAB-PPR returns an $\epsilon$-optimal arm, and its cost is upper bounded by*

$$O\left(W \log(1/\delta) \cdot \max\left(\frac{1}{\epsilon^2}, \frac{V}{\epsilon}\right)\right).$$

Here and below, $O(\cdot)$ stands for universal multiplicative constants. In comparison, previous algorithms for the unit-cost model, such as the ones studied in [14], would cost here at least $\Omega(W \log(K/\delta)/\epsilon^2)$ where $K$ is the number of arms, assuming bounded rewards. Qualitatively, removing the dependence on $K$ is of significance. MAB-PPR obtains a quantitatively significant improvement when $K$ is large and $V \leq O(1/\epsilon)$.

**Algorithm 1** MAB-PPR: $\epsilon$-Best-Arm-Identification with Pay-Per-Reward
___
**Input:** $\epsilon, \delta \in (0,1)$, $V \geq 1$, $K \in \mathbb{N}$; Universal constants $C_q, C_q', C_1, C_2, C_3, N_{\text{final}} > 0$; $\rho, \theta \in$
    $(0,1)$.
**Output:** $r \in [K]$
 1: $t \leftarrow 1, S_1 \leftarrow [K], \epsilon_1 \leftarrow (1-\rho)\epsilon/2, \delta_1 \leftarrow \delta/2$.
 2: **while** $|S_t| > N_{\text{final}}$ **do**
 3:     ▷ *First batch of pulls:*
 4:     Set $q_1 \leftarrow C_q \log(C_q'/\delta_t)$ and $\ell_t \leftarrow C_1 \max(1/\epsilon_t^2, V/\epsilon_t)$; Pull the arms in $S_t$ for $q_1 \ell_t$ times.
 5:     For each $i \in S_t$, use the pulls of arm $i$ to calculate $\hat{w}_{i,t}$, the $(q_1, \ell_t)$-MoM estimate of $w_i$.
 6:     Set $\{j_l\}_{l \in [|S_t|]}$ to be an ordering of $S_t$ such that $\hat{w}_{j_l,t} \leq \hat{w}_{j_{l+1},t}$ for all $l \in [|S_t| - 1]$.
 7:     ▷ *Second batch of pulls:*
 8:     Set $q_2 \leftarrow \frac{9}{2} \log(12/\delta_t)$, $\ell_{(2)} \leftarrow C_2 V/\epsilon$; Pull the arms in $S_t$ for $q_2 \ell_{(2)}$ times.
 9:     For $i \in S_t$, let $\{x_i^{j,l}\}_{j \in [q_2], l \in [\ell_{(2)}]}$ be the i.i.d. copies of $X_i$ obtained in line 8.
10:     **for** $l \in [|S_t|]$ **do**
11:         Define $A_l := \{j_1, \ldots, j_l\}$.
12:         Use $\{\sum_{i \in A_l} x_i^{j,l}\}_{j \in [q_2], l \in [\ell_{(2)}]}$ to calculate $\widetilde{W}_{A_l,t}$, the $(q_2, \ell_{(2)})$-MoM estimate of $W_{A_l}$.
13:     **end for**
14:     ▷ *Decide which arms to remove:*
15:     If $\widetilde{W}_{S_t,t} \leq \epsilon/4$, **return** some $r \in S_t$ and terminate.
16:     Find the largest integer $N_t$ such that $\widetilde{W}_{A_{N_t},t}/\widetilde{W}_{S_t,t} \leq \theta$.
17:     Set $S_{t+1} \leftarrow S_t \setminus A_{N_t}$, $\epsilon_{t+1} \leftarrow \rho\epsilon_t$, $\delta_{t+1} \leftarrow \delta_t/2$, $t \leftarrow t+1$.
18: **end while**
19: ▷ *Final round ($|S_t| \leq N_{\text{final}}$):*
20: Set $q_3 \leftarrow \frac{9}{2} \log(N_{\text{final}}/\delta_t)$, $\ell_{(3)} \leftarrow C_3/\epsilon^2$; Pull the arms in $S_t$ for $q_3 \ell_{(3)}$ times.
21: For $i \in S_t$, use the pulls of arm $i$ to calculate $\hat{w}_{i,t}$, the $(q_3, \ell_{(3)})$-MoM estimate of $w_i$.
22: **return** some $r \in \operatorname{argmax}_{i \in S_t} \hat{w}_{i,t}$.
___

## 4   The **MAB-PPR** algorithm

The MAB-PPR algorithm is listed in Alg. 1. The main ideas in MAB-PPR are described below. We use several universal constants in Alg. 1. In the analysis below, we show that there exist values for these constants such that the guarantees of MAB-PPR hold. For readability, we do not explicitly round up non-integer sample sizes. Such rounding would not affect our final cost upper bound.

A central challenge in designing MAB-PPR is avoiding a union bound on the number of arms. In [14], where arm pulls have unit cost, the median-elimination algorithm was proposed for this purpose. This algorithm removes half of the arms from the candidate set in each round. In our case, because of the different cost model, it is necessary to remove a constant fraction of the total weight, regardless of the number of arms. To achieve this, MAB-PPR uses two batches of arm pulls in each round. The universal constants $C_q, C_q', C_1, C_2$ are used for calculating the required numbers of arm pulls in each batch so that subsequent estimates hold. The first batch of pulls (line 4) is used to estimate the weight of each arm with the MoM estimator (line 5). The arms are then ordered according to their estimated weight. The second batch of pulls (line 8) is used to decide how many arms to remove from the bottom of this ordering, to make sure that the removed fraction of the total weight is within a certain range. This is done by estimating the total weight of each possible subset of arms, again using MoM (line 12). Here, it is important to note that taking the sum of MoM estimates is not equivalent to taking the MoM estimate of the sum, thus these sums are estimated directly. The selected number of arms is then removed from the bottom of the ordering (line 16). The threshold $\theta$ is used to upper bound the empirical fraction of the removed arms. After each round, the values of $\epsilon_t$ and $\delta_t$ are updated, where the constant $\rho$ controls the rate of decay for $\epsilon_t$. MAB-PPR terminates in one of two ways: if at some point, the estimated total weight of the remaining arms is small, then some remaining arm is returned (line 15). Otherwise, when the number of remaining arms is smaller than the constant $N_{\text{final}}$, all their weights are directly estimated (where the constant $C_3$ is used to set the number of pulls), and the arm with the maximal estimated weight is returned (line 22).

**Supporting an unbounded number of arms.** For simplicity of presentation, the notations and Alg. 1 are given for a finite and known number of arms $K$. Nonetheless, MAB-PPR can be used even if the number of arms is unbounded or infinite. This is possible if there is a way to pull all arms together in finite time, and get a finite result listing all the non-zero instantaneous rewards. For instance, in the search-query example, pulling all arms together can be done by placing an ad and linking it to all search queries, and then recording which search queries actually came up and resulted in an ad-click. Assuming that $W \equiv \sum_i w_i < \infty$, Alg. 1 can then support an unbounded number of arms as follows. In line 1, $S_1$ should be (implicitly) set to the entire set of available arms. In line 6 of the first round, only arms with a non-zero reward in any of the pulls of the first batch need to be explicitly ordered. The other arms can be implicitly placed at the bottom of the ordering. After the second batch, let $l_0$ be the smallest index such that arm $j_{l_0}$ got a positive reward at least once. Then for all $l < l_0$, $\widetilde{W}_{A_l,1}$ can be (implicitly) set to zero. Clearly, the value of $N_1$, set in line 16 of the first round, satisfies $N_1 \geq l_0$. Thus, in the first round, any arm that got no reward in any of the pulls is removed in line 17. As a result, the set of arms $S_2$ set for the second round is finite and known, and so from this point on, the algorithm can proceed as listed.

# 5 Analysis

In this section we prove Theorem 3.1, restated below in more detail as Theorem 5.6. Recall that $S_t$ is the set of candidate arms in round $t$. Denote by $W_t := W_{S_t}$ the total weight of the arms in $S_t$. As in Alg. 1, the estimations made by MAB-PPR are denoted by $\{\hat{w}_{i,t}\}_{i \in S_t}$ and $\{\widetilde{W}_{A_l,t}\}_{l \in [|S_t|]}$. We omit the subscript $t$ when it is clear from context. Denote $\widetilde{W}_t := \widetilde{W}_{S_t,t}$. We say that a round $t$ of MAB-PPR is *standard* if $|S_t| > N_{\text{final}}$ and $W_t \geq \epsilon/8$. We first show that with a high probability, the only non-standard round (if at all) is the last round. Define the following event:

$$E_0(t) := \{(W_t < \epsilon/8 \Rightarrow \widetilde{W}_t \leq \epsilon/4) \text{ and } (W_t \geq \epsilon/2 \Rightarrow \widetilde{W}_t \geq \epsilon/4)\}.$$

**Lemma 5.1.** *For a large enough constant $C_2$, for any round $t$ with $|S_t| > N_{\text{final}}$, $E_0(t)$ holds with a probability at least $1 - \delta_t/4$.*

*Proof.* If $|S_t| > N_{\text{final}}$ then line 8 is invoked, and $\widetilde{W}_t$ is a $(q_2, \ell_{(2)})$-MoM estimate of $W_t$. By Assumption 2.2, $\sigma_{S_t}^2 \leq V W_t$. Hence, by Lemma 2.3, since $q_2 \geq \frac{9}{2} \log(4/\delta_t)$, with a probability at least $1 - \delta_t/4$, $|W_t - \widetilde{W}_t| \leq \sqrt{6 V W_t/\ell_{(2)}}$. We have $\ell_{(2)} = C_2 V/\epsilon$. By setting $C_2 \geq 48$, we get $|W_t - \widetilde{W}_t| \leq \sqrt{W_t \epsilon/8}$. If $W_t < \epsilon/8$, it follows that $\widetilde{W}_t \leq W_t + \epsilon/8 \leq \epsilon/4$, which shows that the first part of $E_0(t)$ holds. If $W_t \geq \epsilon/2$, then $\widetilde{W}_t \geq W_t - \sqrt{W_t \epsilon/8} \geq W_t/2 \geq \epsilon/4$, which shows that the second part of $E_0(t)$ holds. $\qquad\square$

If $E_0(t)$ holds and $W_t < \epsilon/8$, then in round $t$ the condition in line 15 is invoked, leading to the termination of MAB-PPR. The other option for a round $t$ to be non-standard is if $|S_t| \leq N_{\text{final}}$. This case also leads to the termination of the algorithm. Therefore, if $E_0(t)$ holds for all rounds, then all rounds except for (perhaps) the last one are standard. We use this observation in the proofs below.

We prove the correctness of MAB-PPR by guaranteeing that with a high probability, an $\epsilon$-best arm remains in the set $S_t$ throughout the algorithm. We upper bound the cost of MAB-PPR by showing that under the same events, the total weight of arms is reduced in every round by at least a certain fraction. This general analysis structure is also used for the median-elimination algorithm in [14], where half of the arms are removed in each round. However, once weights are used instead of numbers of arms, new techniques are required, and the estimation scheme, as well as the analysis, become more involved.

First, we define several events. We later prove that they all hold together with a high probability. Let $i_t^* \in \arg\max_{i \in S_t} w_i$ be an arm with the largest weight in $S_t$. Let $w_t^* := w_{i_t^*}$, and $\hat{w}_t^* := \hat{w}_{i_t^*,t}$. Note that $i^* := i_1^*$ is the optimal arm and its weight is $w^* := w_1^*$. For any arm $j \in S_t$, we say that $j$ is a *t-bad* arm if $w_j < w_t^* - \epsilon_t$, where $\epsilon_t$ is as defined in Alg. 1. We also define the set $M_t := \{i \in S_t \mid i \text{ is } t\text{-bad and } \hat{w}_{i,t} \geq \hat{w}_t^*\}$.

We define three constants: $\phi_L, \phi_M, \phi_U \in (0,1)$ (standing for "Lower", "Middle" and "Upper"), such that $\phi_L < \phi_M < 1 - \phi_U$. These constants are used to analyze the fraction of removed arms in

each round, as follows: $\phi_L$ is a lower-bound on the fraction of the arm weight which is guaranteed to be removed in each round. $\phi_U$ is an upper bound on the fraction of the arm weight in bad arms. $\phi_M$ is used to define a set of arms guaranteed to be removed in each round. Denote the set of arms that appear to be the worst in round $t$ according to the estimates $\{\hat{w}_{i,t}\}$ by $L_t = \{j_1, \ldots, j_{k_t}\}$, where $\{j_l\}_{l \in [|S_t|]}$ is the ordering defined in line 6 of round $t$, and $k_t$ is the largest integer such that $\sum_{i \in L_t} w_i \leq \phi_M W_t$. We define the following events for each round $t$, where $\phi_L, \phi_U \in (0,1)$ are constants such that :

- $E_1(t) := \{W_{M_t} < \phi_U W_t\}$,
- $E_2(t) := \{W_{L_t} \geq \phi_L W_t \text{ or } |S_t \setminus L_t| \leq N_{\text{final}}\}$,
- $E_3(t) := \{(S_{t+1} \setminus M_t \neq \emptyset) \wedge (S_{t+1} \cap L_t = \emptyset)$.

We omit the argument $(t)$ on events when it is clear from context. The following lemmas state that in a standard round $t$, all of these events hold with a high probability, as long as the constants are selected appropriately. Their proofs are provided after the statement and proof of Theorem 5.6.

**Lemma 5.2.** *For any $\phi_U \in (0,1)$ and sufficiently large constants $C_q, C_q', C_1 > 0$, we have that for any standard round $t$, with a probability at least $1 - \delta_t/4$, $E_1(t)$ holds.*

**Lemma 5.3.** *For any $\phi_L, \phi_M, \phi_U$ such that $0 < \phi_L < \phi_M < 1 - \phi_U < 1$, there are sufficiently large constants $C_q, C_q', C_1, N_{\text{final}} > 0$ such that for any standard round $t$, with a probability at least $1 - \delta_t/4$, $E_2(t)$ holds.*

**Lemma 5.4.** *For any $\phi_L, \phi_M, \phi_U$ such that $0 < \phi_L < \phi_M < 1 - \phi_U < 1$, there is some $\theta \in (0,1)$ such that for a sufficiently large $C_2 > 0$, in any standard round $t$, with a probability at least $1 - \delta_t/4$, $E_1(t)$ implies $E_3(t)$.*

The following lemma shows that these events guarantee that a good arm always remains in the candidate set, and that the reduction of the candidate set in each round is as required.

**Lemma 5.5.** *For any standard round $t$ in which $E_1, E_2, E_3$ all hold, we have, with a probability at least $1 - \delta_t$, that (1) $w_{t+1}^* \geq w_t^* - \epsilon_t$, and (2) $W_{t+1} \leq (1 - \phi_L)W_t$ or $|S_{t+1}| \leq N_{\text{final}}$.*

*Proof.* To prove the first part, observe that by $E_3$, $S_{t+1} \setminus M_t \neq \emptyset$. Let $j \in \text{argmax}_{j \in S_{t+1} \setminus M_t} \hat{w}_{j,t}$. By the definition of $S_{t+1}$ in Alg. 1, it includes the arms in $S_t$ with the largest estimated weights. Thus, we also have $j \in \text{argmax}_{j \in S_t \setminus M_t} \hat{w}_{j,t}$. Therefore, since $i_t^* \in S_t \setminus M_t$, it follows that $\hat{w}_{j,t} \geq \hat{w}_t^*$. Thus, since $j \notin M_t$, it follows $j$ is not $t$-bad, so $w_j \geq w_t^* - \epsilon_t$. Combined with $w_{t+1}^* \geq w_j$, this completes the proof of the first part. For the second part, note that $S_{t+1} \subseteq S_t$ and $L_t \subseteq S_t$. By $E_3$, $S_{t+1} \cap L_t = \emptyset$. Therefore, $W_{t+1} + W_{L_t} \leq W_t$. If $|S_{t+1}| > N_{\text{final}}$ then by $E_2$, $W_{L_t} \geq \phi_L W_t$. Therefore, $W_{t+1} \leq W_t(1 - \phi_L)$. $\square$

Combining the lemmas above, the main result is shown in the following theorem.

**Theorem 5.6.** *For any setting of the constants except for $C_3$ and $\rho$ that satisfies the lemmas above, there is setting of $C_3 > 0$ and $\rho \in (0,1)$ such that with a probability at least $1 - \delta$, MAB-PPR terminates and returns an $\epsilon$-best arm, and its cost is*

$$O(W \cdot \log(1/\delta) \cdot \max(\frac{1}{\epsilon^2}, \frac{V}{\epsilon})).$$

*Proof.* Condition below on $E_0$ occurring in all rounds, and on $E_1, E_2, E_3$ occurring in all standard rounds. By Lemmas 5.1, 5.2, 5.3, and 5.4, and the fact that $\sum_{t=1}^{\infty} \delta_t \leq \delta$, this condition holds with a probability at least $1 - \delta$. By Lemma 5.5, in all standard rounds $W_t \leq (1 - \phi_L)^{t-1}W$. From $E_0(t)$ it follows that in a round with $W_t \leq \epsilon/8$, we have $\widetilde{W}_t \leq \epsilon/4$, which guarantees that MAB-PPR terminates in line 15. Therefore, MAB-PPR terminates at the latest when $t$ satisfies $(1 - \phi_L)^{t-1}W \leq \epsilon/8$. Let $T$ be the last round of the run. By Lemma 5.5, in standard rounds we have $w_{t+1}^* \geq w_t^* - \epsilon_t$. Thus,

$$w_T^* \geq w^* - \sum_{t=1}^{T-1} \epsilon_t = w^* - (1 - \rho) \sum_{t=1}^{T-1} \rho^{t-1}\epsilon/2 \geq w^* - \epsilon/2. \tag{1}$$

Consider the two possible ways of terminating: if the algorithm terminates in line 15, then $\widetilde{W}_T \leq \epsilon/4$. By $E_0(t)$, this means that $W_T \leq \epsilon/2$, thus $w_r \geq w_T^* - \epsilon/2$. The other way of terminating is when $|S_T| \leq N_{\text{final}}$. In this case, for all $i \in S_T$, $\hat{w}_{i,T}$ is a $(q_3, \ell_{(3)})$-MoM estimate of $w_i$, where $q_3 \geq \frac{9}{2} \log(N_{\text{final}}/\delta_T)$ and $\ell_{(3)} = C_3/\epsilon^2$. Set $C_3 \geq 96$. By Lemma 2.3 and a union bound on $N_{\text{final}}$ arms, it follows that with a probability at least $1 - \delta_T$, $\forall i \in S_T, |w_i - \hat{w}_{i,T}| \leq \sqrt{6\sigma_i^2/\ell_{(3)}} \leq \sqrt{6/\ell_{(3)}} \leq \epsilon/4$, where the second inequality follows from Assumption 2.1. The returned arm in this case is $r \in \operatorname{argmax}_{i \in S_T} \hat{w}_{i,T}$, thus $w_r \geq \hat{w}_{r,T} - \epsilon/4 \geq \hat{w}_T^* - \epsilon/4 \geq w_T^* - \epsilon/2$. Therefore, in both cases we have $w_r \geq w_T^* - \epsilon/2$. Combined with Eq. (1), this proves that $w_r \geq w^* - \epsilon$.

The cost of any round of MAB-PPR is upper-bounded by $W_t(q_1\ell_t + q_2\ell_{(2)} + q + 3\ell_{(3)})$. Therefore, the total cost is at most $W \sum_{t=1}^{T}(1 - \phi_L)^{t-1}(q_1\ell_t + q_2\ell_{(2)} + q_3\ell_{(3)})$. We have, for constants $C, C' > 0$ (that depend on the constants used in Alg. 1), that $q_1\ell_t + q_2\ell_{(2)} + q_3\ell_{(3)} \leq C \log(\frac{C'}{\delta_t}) \cdot \max(\frac{1}{\epsilon_t^2}, \frac{V}{\epsilon_t})$, where $\delta_t = \delta/2^t$ and $\epsilon_t = \frac{1}{2}(1 - \rho)\rho^{t-1}\epsilon$. Therefore, the total cost is upper bounded by

$$W \sum_{t=1}^{T}(1 - \phi_L)^{t-1} \cdot C \log(\frac{C'}{\delta_t}) \max(\frac{1}{\epsilon_t^2}, \frac{V}{\epsilon_t}) \leq C'' \cdot W \log(C'''/\delta) \max(\frac{1}{\epsilon^2}, \frac{V}{\epsilon}) \sum_{t=1}^{T-1} \frac{t(1 - \phi_L)^{t-1}}{(\rho^2)^{t-1}},$$

For constants $C'', C''' > 0$. Setting $\rho \in (\sqrt{1 - \phi_L}, 1)$, the sum on $t$ is upper-bounded by a constant, giving the required upper bound on the cost. $\square$

We now prove lemmas that the events $E_1, E_2, E_3$ hold with a high probability. We start with Lemma 5.2, which shows this for $E_1(t)$, which states that $W_{M_t} < \phi_U W_t$.

*Proof of Lemma 5.2.* Fix a round $t$. Define the event $E_4 := \{\hat{w}_t^* \geq w_t^* - \epsilon_t/2\}$. Setting $C_q \geq \frac{9}{2}$, $C'_q \geq 8$ and $C_1 \geq 24$, we have $q_1 \geq \frac{9}{2}\log(8/\delta_t)$ and $\ell_t \geq 24/\epsilon_t^2$. By Assumption 2.1, $\sigma_{i_t^*}^2 \leq 1$. Hence, by Lemma 2.3, with a probability $1 - \delta_t/8$, $|w_t - \hat{w}_t^*| \leq \sqrt{6/\ell_t} \leq \epsilon_t/2$, hence $E_4$ holds.

Let $i \in S_t$ be a $t$-bad arm. Let $\hat{w}_i^j$ for $j \in [q_1]$ be the $j$'th mean estimate of $w_i$ used to get the estimate $\hat{w}_{i,t}$ in line 5. Let $\alpha_1 \in (1 - \frac{\phi_U}{2}, 1)$. Set $\alpha_2 = 1 - \frac{1-\phi_U/2}{\alpha_1}$, and set $\alpha_3 \in (0, \alpha_2(1 - \alpha_1))$. If $E_4$ hold, then since $i$ is $t$-bad, $\hat{w}_i^j \geq \hat{w}_t^*$ implies $\hat{w}_i^j \geq \hat{w}_t^* \geq w_i + \epsilon_t/2$. Denote $\mathbb{P}_t[\cdot] := \mathbb{P}[\cdot \mid S_t]$. Then,

$$\mathbb{P}_t[\hat{w}_i^j \geq \hat{w}_t^* \mid E_4] \leq \mathbb{P}_t[\hat{w}_i^j \geq w_i + \epsilon_t/2]/\mathbb{P}_t[E_4] \leq 2\mathbb{P}_t[\hat{w}_i^j \geq w_i + \epsilon_t/2] \leq \frac{2}{\ell_t(\epsilon_t/2)^2}.$$

The last inequality follows from Chebychev's inequality, since $\sigma_i^2 \leq 1$. Setting $C_1 \geq \frac{8}{\alpha_3}$, we have $\ell_t \geq \frac{8}{\alpha_3\epsilon_t^2}$, hence $\mathbb{P}_t[\hat{w}_i^j \geq \hat{w}_t^* \mid E_4] \leq \alpha_3$. Denote $M_t^j := \{i \mid i \text{ is } t\text{-bad and } \hat{w}_i^j \geq \hat{w}_t^*\}$. We conclude that

$$\mathbb{E}_t[W_{M_t^j} \mid E_4] = \sum_{i \in S_t} w_i \cdot \mathbb{P}_t[i \in M_t^j \mid E_4] \leq \sum_{i \in S_t} w_i \cdot \mathbb{P}_t[\hat{w}_i^j \geq \hat{w}_t^* \mid E_4] \leq \alpha_3 W_t.$$

By Markov's inequality, it follows that $\mathbb{P}_t[W_{M_t^j} \geq \alpha_2 W_t \mid E_4] \leq \frac{\alpha_3 W_t}{\alpha_2 W_t} < 1 - \alpha_1$. Define the event $E_5 := \{\frac{1}{q_1}\sum_j \mathbb{I}[W_{M_t^j} \geq \alpha_2 W_t] < 1 - \alpha_1\}$. We have $q_1 \geq C_q \log(C'_q/\delta_t)$. Setting $C_q \geq 1/(2(1 - \alpha_1 - \alpha_3/\alpha_2)^2)$ and $C'_q \geq 8$, we have by Hoeffding's inequality and the independence of $\{M_i^j\}$ for $j \in [q_1]$ that $\mathbb{P}_t[E_5 \mid E_4] \geq 1 - \delta_t/8$. Therefore, combined with $\mathbb{P}_t[E_4] \geq 1 - \delta_t/8$, we get that for any $S_t$, $\mathbb{P}_t[E_5] \geq 1 - \delta/4$. Under $E_5$, we have

$$\frac{1}{q_1}\sum_j W_{M_t^j} < \alpha_2\alpha_1 W_t + (1 - \alpha_1)W_t = (\alpha_2\alpha_1 + 1 - \alpha_1)W_t = \phi_U W_t/2. \tag{2}$$

Where the last equality follows from the definition of $\alpha_2$. On the other hand, since $\hat{w}_{i,t}$ is the median of $\{\hat{w}_i^j\}_{j \in [q_1]}$, and $i \in M_t$ implies $\hat{w}_{i,t} \geq \hat{w}_t^*$, we have

$$W_{M_t} = \sum_{i \in S_t} w_i \cdot \mathbb{I}[i \in M_t] \leq \sum_{i \in S_t} w_i \cdot \mathbb{I}[|\{j \mid i \in M_t^j\}| \geq q_1/2]$$

$$\leq \frac{2}{q_1}\sum_{i \in S_t} w_i \sum_{j \in [q_1]} \mathbb{I}[i \in M_t^j] = \frac{2}{q_1}\sum_{j \in [q_1]}\sum_{i \in S_t} w_i \cdot \mathbb{I}[i \in M_t^j] = \frac{2}{q_1}\sum_{j \in [q_1]} W_{M_t^j}.$$

Combined with Eq. (2), we get that with a probability of $1 - \delta_t/4$, $W_{M_t} < \phi_U W_t$. $\square$

We now prove Lemma 5.3, which states that with a probability at least $1 - \delta_t/4$, $E_2(t)$ holds in a standard round, i.e., $W_{L_t} \geq \phi_L W_t$ or $|S_t \setminus L_t| \leq N_{\text{final}}$. In this proof we use the following lemma, proved in the supplementary material (Appendix C).

**Lemma 5.7.** *Let $Y_1, \ldots, Y_z$ be random variables with means $\mu_1, \ldots, \mu_z$. Let $\bar{Y} := \sum_{i \in [z]} Y_i$, $\mu := \sum_{i \in [z]} \mu_i = \mathbb{E}[\bar{Y}]$. Let $\sigma^2 := \text{Var}(\bar{Y}) < \infty$. For $i \in [z]$, let $\hat{\mu}_i$ be the $(q, \ell)$-MoM estimate of $\mu_i$ based on $q\ell$ i.i.d. draws of copies of $Y_i$. Then for any $\gamma > 0$, with a probability at least $1 - \exp(-q/18)$, $|\{i \in [z] \mid \hat{\mu}_i \geq \gamma\mu\}| \leq \frac{4}{\gamma}(1 + \sqrt{\frac{6\sigma^2}{\mu^2\ell}})$.*

*Proof of Lemma 5.3.* We will assume that $W_{L_t} < \phi_L W_t$ and conclude the required upper bound on $|S_t \setminus L_t|$ with a high probability. Let $\beta := \phi_M - \phi_L$. Let $\nu \in (0, \beta)$, and set $\alpha := \frac{1}{2}(\beta - \nu)/\sqrt{\beta}$. Let $j := j_{k_t+1}$, where $k_t$ is given in the definition of $L_t$ and $\{j_l\}$ is the ordering set in line 6 in round $t$. By the definition of $L_t$, $W_{L_t} + w_j > \phi_M W_t$. By our assumption, $W_{L_t} < \phi_L W_t$. Therefore, $w_j > (\phi_M - \phi_L)W_t = \beta W_t$. Define $J = \{i \in S_t \mid w_i \geq \beta W_t\}$. We have $|J| \leq 1/\beta$.

Set $C_q \geq \frac{9}{2}$ and $C'_q \geq 8/\beta$, so that $q_1 \geq \frac{9}{2}\log((8/\beta)/\delta_t)$. Recall that $\sigma_i^2 \leq w_i$ by Assumption 2.1. Therefore, by Lemma 2.3 and a union bound, with a probability at least $1 - \delta_t/8$, $\forall i \in J, |w_i - \hat{w}_i| \leq \sqrt{6w_i/\ell_t}$. Setting $C_1 \geq 6/\alpha^2$, it follows that $\ell_t \geq 6/(\alpha^2\epsilon_t)$. We have $\epsilon_t \leq \epsilon/2$ for all $t$. In addition, since $t$ is a standard round, $\epsilon/8 \leq W_t$. Therefore, for $i \in J$, $\epsilon_t \leq 4W_t \leq 4w_i/\beta$. It follows that with a probability at least $1 - \delta_t/8$, $\forall i \in J, |w_i - \hat{w}_i| \leq (2\alpha/\sqrt{\beta})w_i = (1 - \nu/\beta)w_i$, where the equality follows from the definition of $\alpha$. Hence $\hat{w}_i \geq (\nu/\beta) \cdot w_i \geq \nu W_t$. In particular, this holds for $j = j_{k_t+1}$ as set above. Condition below on this event.

By the definition of $L_t$ and $j$, for all $i \in S_t \setminus L_t$, $\hat{w}_i \geq \hat{w}_j$. Therefore, by the event above, for all $i \in S_t \setminus L_t$, $\hat{w}_i \geq \nu W_t$. Hence, $|S_t \setminus L_t| \leq |\{i \in S_t \mid \hat{w}_i \geq \nu W_t\}|$. To bound the RHS, recall that $\sigma_{S_t}^2 \leq V W_t$ by Assumption 2.2. Thus, applying Lemma 5.7, we have that by setting $C_q \geq 18$, w.p. $1 - \delta_t/8$, $|\{i \in S_t \mid \hat{w}_i \geq \nu W_t\}| \leq \frac{4}{\nu}(1 + \sqrt{\frac{6V}{W_t\ell_t}})$. We have $\ell_t \geq C_1 V/\epsilon_t \geq C_1 V/(4W_t)$. Therefore,

$$|\{i \in S_t \mid \hat{w}_i \geq \nu W_t\}| \leq \frac{4}{\nu}(1 + \sqrt{24/C_1}) =: N_{\text{final}}. \tag{3}$$

By a union bound, with a probability $1 - \delta_t/4$, the event above and Eq. (3) hold simultaneously, in which case $W_{L_t} < \phi_L W_t$ implies $|S_t \setminus L_t| \leq N_{\text{final}}$. $\qquad\square$

Lastly, we prove Lemma 5.4, which states that in a standard round $t$, with a probability at least $1 - \delta_t/4$, $E_1(t)$ implies $E_3(t)$, i.e., $S_{t+1} \setminus M_t \neq \emptyset$ and $S_{t+1} \cap L_t = \emptyset$.

*Proof of Lemma 5.4.* Fix a round $t$. Let $A_l := \{j_1, \ldots, j_l\}$, where $\{j_l\}$ is the ordering set in line 6, and let $\widetilde{W}_{A_l} := \widetilde{W}_{A_l,t}$ be the estimate of $W_{A_l}$ calculated in line 8 of round $t$. By the assumption of the lemma, $\phi_M < 1 - \phi_U$. Let $R \in (0, 1 - \phi_U)$ be the solution to $\frac{\phi_M + R\sqrt{\phi_M}}{1-R} - \frac{1-\phi_U-R}{1+R} = 0$. The solution is in this range, since since for $R = 0$, the LHS is equal to $\phi_M - (1 - \phi_U) < 0$, and for $R = 1 - \phi_U$, the LHS is positive. Set $\theta := \frac{1-\phi_U-R}{1+R}$.

Recall that $q_2 = \frac{9}{2}\log(12/\delta_t)$. By setting $C_2 \geq 48/R^2$, we get $\ell_{(2)} \geq 48V/(R^2\epsilon)$. Therefore, by Lemma 2.3 and Assumption 2.2, for any fixed $l$ the following holds with a probability $1 - \delta_t/12$:

$$|\widetilde{W}_{A_l} - W_{A_l}| \leq \sqrt{6VW_{A_l}/\ell_{(2)}} \leq \sqrt{R^2 W_{A_l}\epsilon/8} \leq R\sqrt{W_{A_l}W_t}. \tag{4}$$

The last inequality follows since $t$ is a standard round, hence $W_t \geq \epsilon/8$. Let $n_t$ be the smallest integer such that $W_{A_{n_t}} \geq W_t - W_{M_t}$. With a probability $1 - \delta_t/4$, Eq. (4) holds simultaneously for $l = n_t, l = k_t$ (so $A_l \equiv L_t$) and $l = |S_t|$ (so $A_l \equiv S_t$). Condition below on this combined event. It follows from the definition of $L_t$ that

$$\widetilde{W}_{L_t} \leq W_{L_t} + R\sqrt{W_{L_t}W_t} \leq (\phi_M + R\sqrt{\phi_M})W_t.$$

Suppose that $E_1(t)$ holds, so $W_{M_t} < \phi_U W_t$. By definition of $n_t$, $W_{A_{n_t}} > (1 - \phi_U)W_t$. Therefore,

$$\widetilde{W}_{A_{n_t}} \geq W_{A_{n_t}} - R\sqrt{W_{A_{n_t}}W_t} > (1 - \phi_U)W_t - RW_t = (1 - \phi_U - R)W_t.$$

Lastly, from Eq. (4) for $l = |S_t|$, we get $W_t(1 - R) \leq \widetilde{W}_t \leq W_t(1 + R)$. Now, MAB-PPR sets $S_{t+1} = S_t \setminus A_{N_t}$, where $N_t$ is the largest number that satisfies $\widetilde{W}_{A_{N_t}}/\widetilde{W}_t \leq \theta$. From the inequalities above, we have $\frac{\widetilde{W}_{A_{N_t}}}{\widetilde{W}_t} > \frac{1 - \phi_U - R}{1 + R} = \theta$ and $\frac{\widetilde{W}_{L_t}}{\widetilde{W}_t} \leq \frac{\phi_M + R\sqrt{\phi_M}}{1 - R} = \theta$. Hence, $k_t \leq N_t < n_t$. From $k_t \leq N_t$, we have $S_{t+1} \cap L_t = \emptyset$. From $N_t < n_t$, we have $S_{t+1} \equiv S_t \setminus A_{N_t} \supseteq S_t \setminus A_{n_t - 1}$. Therefore $W_{t+1} \geq W_t - W_{A_{n_t-1}}$. By the definition of $n_t$, we have $W_{A_{n_t-1}} < W_t - W_{M_t}$. Therefore, $W_{t+1} \geq W_{M_t}$. It follows that $S_{t+1} \setminus M_t \neq \emptyset$. Thus, $E_1(t)$ implies $E_3(t)$. $\qquad\square$

Having proved Lemma 5.2,Lemma 5.3, and Lemma 5.4, this finalizes the proof of Theorem 5.6.

## 6  Conclusion

In this work we showed that it is possible to identify an $\epsilon$-best arm in a pay-per-reward multi-armed-bandit setting at a cost that does not depend on the number of arms, and depends only linearly on the total expected reward. In future work, we plan to study instance-specific bounds for this setting.

**Acknowledgements**

Sivan Sabato was supported in part by the Israel Science Foundation (grant No. 555/15).

## Footnotes

[1]We discuss in Section 4 how to use MAB-PPR when the set of available arms is infinite or unknown.

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
