[Supplementary Material]

# $\epsilon$-Best-Arm Identification in Pay-Per-Reward Multi-Armed Bandits
## Supplementary Material

## A  A bound on $V$

We show that if each arm is positively correlated with at most $Q$ other arms, then $V$ in Assumption 2.2 can be set to $1 + 2Q$. For a set of arms $A \subseteq [K]$ and $i \in A$, let $P(i)$ denote the set of arms in $A$ that are different from $i$ and positively correlated with it. Assume w.l.o.g. that $j < i \Rightarrow w_j \le w_i$. Then, using Assumption 2.1,

$$\sigma_A^2 = \sum_{i \in A} \sigma_i^2 + \sum_{i,j \in A, i \ne j} \mathrm{Cov}(X_i, X_j)$$

$$\le W_A + \sum_{i,j \in A, i \ne j} \mathbb{I}[j \in P(i)] \sqrt{\sigma_i^2 \sigma_j^2}$$

$$\le W_A + 2 \sum_{i \in A} \sum_{j \in A, j < i} \mathbb{I}[j \in P(i)] \sqrt{w_i w_j}$$

$$\le W_A + 2 \sum_{i \in A} \sum_{j \in A, j < i} \mathbb{I}[j \in P(i)] w_i$$

$$= W_A + 2 \sum_{i \in A} w_i \sum_{j \in A, j < i} \mathbb{I}[j \in P(i)]$$

$$\le W_A + 2 \sum_{i \in A} w_i Q$$

$$\le (1 + 2Q) W_A.$$

Thus, in this case Assumption 2.2 holds with $V = 1 + 2Q$.

## B  Median-of-means estimation

In Section 2, we defined the median-of-means as the median of empirical means of several batches of i.i.d. samples. More generally, one can consider the $(q, \ell)$-$p$-of-means estimate, for $p \in (0, 1)$, in which instead of the median, the estimator is set to a number such that a $p$ fraction of the $q$ empirical means are no larger than it and $1 - p$ fraction are no smaller than it. The $(q, \ell)$-MoM estimate is thus also the $(q, \ell)$-$\frac{1}{2}$-of-means estimate. Lemma B.1 and its proof are similar to those provided in [2, 16], and are brought here for completeness. Lemma 2.3 is obtained from Lemma B.1 by setting $p = \frac{1}{2}$ and $a = 6$. Lemma B.1 is used also in the proof of Lemma 5.7 in Appendix C.

**Lemma B.1** (Median-of-Means). *Let $p \in (0, \frac{1}{2})$. Let $X$ be a random variable with mean $w$ and variance $\sigma^2 < \infty$. Let $q, \ell$ be integers, and let $\hat{w}$ be the $(q, \ell)$-$p$-of-means estimate of $w$ from a sample of $q\ell$ i.i.d. draws of copies of $X$. Let $a > 1/p$. With a probability at least $1 - \exp(-q \cdot 2(p - 1/a)^2)$, $|w - \hat{w}| \le \sqrt{a\sigma^2/\ell}$.*

*Proof.* For $j \in [q]$, let $\hat{w}^j$ be the $j$'th empirical mean used for the estimator. By Chebychev's inequality, $\mathbb{P}[|w - \hat{w}| \le \sqrt{a\sigma^2/\ell}] \ge 1 - 1/a$. For $j \in [q]$, let $b_j = \mathbb{I}[|w - \hat{w}^j| \le \sqrt{a\sigma^2/\ell}]$. Then $\mathbb{E}[b_j] \ge 1 - 1/a$, and $b_1, \dots, b_q$ are independent random variables. By Hoeffding's inequality,

$$\mathbb{P}[\frac{1}{q} \sum_{i=1}^{q} b_i > (1 - p)] \ge \mathbb{P}[(\mathbb{E}[b_i] - \frac{1}{q} \sum_{i=1}^{q} b_i) < p - 1/a] \ge 1 - \exp(-q \cdot 2(p - 1/a)^2).$$

If $\frac{1}{q} \sum_{i=1}^{q} b_i > 1 - p$ then more than $1 - p$ fraction of the empirical means satisfy $|w - \hat{w}^j| \le \sqrt{a\sigma^2/\ell}$. We now show that this holds also for $w$ which is the $p$-of-means estimate. Assume for contradiction that it does not hold for $w$. If $w \le \hat{w}$, it follows that all the $p$-fraction smaller estimates also do not

satisfy the inequality, which is a contrdacition. If $w \geq \hat{w}$ and does not satisfy the inequality, then $w$ is larger than at least $1 - p$ fraction of the estimates. But this would imply $p \geq 1 - p$, which is impossible since $p \leq \frac{1}{2}$. Thus, the $p$-of-means satisfies the inequality, which concludes the proof. $\square$

## C  Proof of Lemma 5.7

*Proof of Lemma 5.7.* For $i \in [z]$, let $\{Y_i^{j,l}\}_{j \in [q], l \in [\ell]}$ be the set of i.i.d. samples of copies of $Y_i$ used to get the MoM estimate $\hat{\mu}_i$. Denote $\hat{\mu}_i^j := \frac{1}{\ell} \sum_{l=1}^{\ell} Y_i^{j,l}$, so that $\hat{\mu}_i$ is the median of $\{\hat{\mu}_i^j\}_{j \in [q]}$. Let $\bar{Y}^{j,l} = \sum_{i \in [z]} Y_i^{j,l}$. Then $\{\bar{Y}^{j,l}\}_{j \in [q], l \in [\ell]}$ is a set of i.i.d. samples of copies of $\bar{Y}$. Let $\hat{\mu}$ be the $(q, \ell)$-1/3-of-means estimate of $\bar{\mu} = \mathbb{E}[\bar{Y}]$ based on these samples. Each mean in this estimator is equal to $\frac{1}{\ell} \sum_{l \in [\ell]} \bar{Y}^{j,l} = \sum_{i \in [z]} \hat{\mu}_i^j$ for some $j \in [q]$. Then $\hat{\mu}$ is at least as large as a 1/3 of the numbers in the set $\{\sum_{i \in [z]} \hat{\mu}_i^j\}_{j \in [q]}$.

By Lemma B.1 with $p = 1/3$ and $a = 6$, with a probability at least $1 - \exp(-q \cdot 2(1/3 - 1/6)^2) = 1 - \exp(-q/18)$,

$$\hat{\mu} \leq \mu + \sqrt{6\sigma^2/\ell}. \tag{5}$$

Now, let $I := \{i \in [z] \mid \hat{\mu}_i \geq \gamma\mu\}$, and $T := |I|$. We have $\forall i \in I, \sum_{j \in [q]} \mathbb{I}[\hat{\mu}_i^j \geq \gamma\mu] \geq q/2$. Hence,

$$\frac{1}{q} \sum_{j \in [q]} \sum_{i \in I} \mathbb{I}[\hat{\mu}_i^j \geq \gamma\mu] \geq T/2.$$

Let $N_j := \sum_{i \in I} \mathbb{I}[\hat{\mu}_i^j \geq \gamma\mu] \leq T$. Letting $J$ be an integer random variable drawn uniformly from $[q]$, we conclude that $\mathbb{E}[N_J] \geq T/2$, so $\mathbb{E}[T - N_j] < T/2$. By Markov's inequality,

$$\mathbb{P}[T - N_J \geq 3T/4] < (T/2)/(3T/4) = 2/3,$$

so $\mathbb{P}[N_J > T/4] \geq 1/3$. It follows that for at least a third of the indices $j \in [q]$, $\{\hat{\mu}_i^j\}_{i \in [z]}$ includes at least $T/4$ values larger than $\gamma\mu$, hence for these indices, $\sum_{i \in [z]} \hat{\mu}_i^j \geq T\gamma\mu/4$. Therefore $\hat{\mu} \geq T\gamma\mu/4$. Combined with Eq. (5), we have that with a probability at least $1 - \exp(-q/18)$,

$$T\gamma\mu/4 \leq \hat{\mu} \leq \mu + \sqrt{6\sigma^2/\ell}.$$

Therefore, $T \leq \frac{4}{\gamma}(1 + \sqrt{\frac{6\sigma^2}{\mu^2\ell}})$, which concludes the proof. $\square$