[Reviews · NeurIPS 2019]

Reviewer 1



This paper introduces a new model for the multi-armed bandit problem, where the cost of pulling any given arm equals the expected utility of that arm (called the pay-per-reward model). The goal is to find an arm whose expected utility is within epsilon of the best arm. An algorithm is given, and it is proved that its regret is bounded by a polylogarithmic factor times the total utilities of all arms. My main objection to this paper is the motivation of this model, which seems somewhat contrived to me. A possible application to ad placement on web is presented, but it's not convincing: it is said that the best arm corresponds to the best expression for an ad, and pulling arms is equivalent to observing the number of clicks when the ad is linked to that expression. But then, the player cannot actively "pull an arm" in any rounds she wants. Also, why is the cost of pulling an arm (which means observing something) equals the reward of the arm? I think of this model as a model that's nice from a mathematical point of view, the result of this paper is non-trivial and the techniques used in the paper to prove the regret bounds are deep, however, I am not sure if it is of broad interest to the NeurIPS community. Also, the regret of this paper is O(W log(1/delta)/eps^2), while in Line 117 they say the regret from previous work [14] is O(W log(K/delta)/eps^2), so I don't call this a "significant" improvement. K is the number of arms. Other comments for authors: - In Line 2 and afterwards, "expected future reward" is ambiguous. Just use "expected reward." - Line 44: the sentence "it is necessary to allow a simultaneous pull of an entire set of arms from the joint distribution of rewards of all arms" is very confusing at this point in the paper. - Line 85: why do you call these "single" arm-pulls, while these are batched arm pulls? - Line 130: step -> line == After reading the response: Thanks for the responses. I now think there may be potential applications for this model, but it needs to be described more clearly in the paper. For the ad placement example, in your model the company is penalized for choosing a good text for the ad, while intuitively it should not be penalized, because choosing a good text means also means more buyers. So, from a game-theoretic point of view, it is still not clear why the company should be penazlied for choosing a good ad. I agree with authors that removing the dependence on the number of arms is significant, and thus I have improved my score.

Reviewer 2



This paper considers the problem where we want to identify an \eps-best-arm in the fixed-confidence multi-armed bandits setting. Unlike the traditional setting, it assumes each arm pull incurs a non-unit cost which equals to the expected future reward of that arm. This setup has potential applications in ad placement: a retailer may want to do some experiments before deciding under which search queries the ad is shown. If a placement incurs a click, the retailer may need to pay for this event. The cost per placement paid to the advertiser is related to the click rate. So the retailer may want to use the minimum cost to identify an \eps-best arm. To achieve this goal, this paper introduces an elimination-based algorithm which is similar to that in [14], but it saves a log(K) factor compared to the uniform sampling or the algorithm straightforwardly derived from [14]. The high-level idea of the proposed algorithm is that instead of removing half arms in each round, it removes arms such that the total weight is decreased by a constant factor with high probability. Because of this the estimation scheme as well as the analysis becomes more involved. Theoretically, this setting is very interesting and the proofs look solid. The paper is well-written. However, I have some concerns about the practical usage of the algorithm. 0) About the motivation that the cost of arm pull is equal to the expected future reward of that arm, I am not sure this is actually the case used in ad placement. I would like to see some concrete real world examples. 1) The authors did not show explicitly the constants in the analysis, which may be very large since the algorithm needs to invoke Median-of-Means procedure repeatedly. While I understand that this is for the convenience of the analysis, large constants may be unaccepted in real world applications despite the fact that the algorithm saves a log(K) factor. 2) I think the fixed-budget setting makes more sense for this problem, since a retailer usually offers a limited budget to do some survey. 3) It would be better if the authors can add some experiments to convince the readers the superior performance of the proposed algorithm. 4) The assumption that "pull all arms in finite time" for manipulating an unlimited number of arms is not practical. Minors: -- It seems that \theta was not used in the proof of Theorem 3.1. However, \theta was specified as a parameter in the main algorithm. Is \theta = \phi_{M}? Please clarify. -- Line 17, “... observes their instantaneous reward”: reward -> rewards -- Line 21. “... arm pull ...”: usage of “arm-pull” should be consistent; there are some other places that this issue exists -- Line 54, “Related work” should be put together with line 55 -- Line 73, “Let K be total the number ...” -> “Let K be the total number ...” -- Line 157, “... in more detail in as Theorem 5.6” -> “ … in more details in Theorem 5.6” -- Line 178, “... is also used in for the median-elimination algorithm ...”: “in” is redundant -- Line 219: "\eps-good arm” -> \eps-best arm -- Appendix A: no “2” factor in the first equality and the first inequality Lemma B.1: p should be at most ½ otherwise the last sentence “... more than 1 - p fraction of the empirical means satisfy \omega - \hat{\omega}^j \leq \sqrt{a \sigma^2 / \ell}, hence the p-of-means estimate also satisfies this inequality” does not hold ------------------------------- [After response] The authors have tried to address the motivation issue. However, I still feel that this problem is a bit artificial, and it is still not clear what real applications this problem will have. Other than this, the technical details of this paper look interesting and the paper is a pleasure to read.

Reviewer 3



Originality: The model, algorithm and analysis seems to be novel, but as a standard generalisation of previous results. Relevant literature seems to be covered. The proofs have not been thoroughly checked, but the arguments and approaches seems to be sound. The heuristic for adapting prior work to the new generalised setting is well founded. Clarity: The paper is generally well written and with a clear focus, comprising of a single result. The non-technical sections are well structured and care seems to be put in conveying the contend. There is however a glaring problem with the technical sections, including the algorithm itself, where more than 10 variables and constants are introduced with little to no explanation, but merely mathematical definition. While this might amount to a functional and correct algorithm, it makes understanding it quite difficult. Unsurprisingly the resulting confusion carries over to the analysis. With this complicated an algorithm, naming quantities so there qualitative function is well motivated would help the clarity tremendously. Significance: The result seems novel and well motivated both in application and literature. It is however incremental, and the scope of the paper is rather small. While keeping the scope well focused is definitely a plus and makes it easy to get an overview of the paper, additional contributions as alluded to in the conclusion might be beneficial. ==== In the rebuttal the authors have promised to address the concerns I raised. I have raised my score slightly due to this.

[Author Response · NeurIPS 2019]

Thank you for the reviews. We address below specific questions/comments (shortened) made by the reviewers.

**Reviewers 1 and 2**

**Question**: Please provide more motivation for the setting.

**Answer**: Applications of the proposed setting include any case where there is an exploration stage in which rewards are
not collected, but payment is still proportional to the quality of the arm. In the paper we give an ad-placement survey
application as an example. As another example, consider a sensor field, where the goal is to find the most active sensor,
but each activity report has a communication cost. Here, "pulling an arm" is done by activating the sensor's reporting
option for a given period of time. Yet another example is that of finding the best crowd worker on a crowd-sourcing
platform. Here, an "arm pull" is equivalent to providing the worker with a test task. The workers are paid according to
their success rate in the test task, thus a more successful worker will be paid more during the exploration.

**Question**: **Reviewer 1**: It is said that the best arm corresponds to the best expression for an ad, and pulling arms is
equivalent to observing the number of clicks when the ad is linked to that expression. But then, the player cannot
actively "pull an arm" in any rounds she wants. Also, why is the cost of pulling an arm (which means observing
something) equals the reward of the arm? **Reviewer 2**: About the motivation that the cost of arm pull is equal to the
expected future reward of that arm, I am not sure this is actually the case used in ad placement.

**Answer**: In the example application, the player can "pull an arm" by linking its ad to the expression represented by this
ad for a given period of time, and then observing the number of clicks. The cost of pulling that arm is thus the number
of clicks on that ad, assuming the common pay-per-click advertisement model.

**Reviewer 1**

**Question**: Why is your regret bound significantly better than those in [14]?

**Answer**: We give an algorithm with a cost that has no dependence on the number of arms. This is a significant
improvement, since it allows any number of arms. For instance, in the ad-placement application, the number of arms
could be the number of search expressions, which is huge. Similarly, there could be a huge number of sensors in a field
of nano sensors. Moreover, completely removing the dependence on the number of arms is of considerable theoretical
significance.

**Question**: Line 85: why do you call these "single" arm-pulls, while these are batched arm pulls?

**Answer**: $n$ is defined to be the total number of times that any arm is pulled, regardless of whether some of the arms are
pulled together or not. We will clarify this.

Thank you for the detailed comments, we will fix all of them.

**Reviewer 2**

**Question**: The authors did not show explicitly the constants in the analysis.

**Answer**: This work shows that is is possible to have no dependence on $K$, which is an interesting and non-trivial result.
We plan to optimize the constants and derive a practical implementation in future work.

**Question**: The assumption that it is possible to "pull all arms in finite time" for manipulating an unlimited number of
arms is not practical.

**Answer**: We explain in lines 141-165 how it can be practical in some applications to pull many or all arms together.
For instance, in the ad-placement example, one can link the ad to all search queries, and simply record which search
queries actually came up and resulted in an ad-click. In the sensor field example, all arms can be "pulled" at the same
time, by activating the reporting option of all sensors at once. In the crowd-worker option, a test task can be uploaded
without any filters so that all crowd-workers can try it.

**Question**: It seems that $\theta$ was not used in the proof of Theorem 3.1.

**Answer**: $\theta$ is used in lemma 5.4. The lemma is then used in the proof of theorem 5.6 (the restatement of theorem 3.1).

Thank you for the detailed comments, we will fix all of them.

**Reviewer 3**

**Question**: Naming quantities so their qualitative function is well motivated would help. Please provide more explana-
tions in the analysis.

**Answer**: We will rename the constants and add explanations on their meaning for better readability. We will add further
explanations to the technical analysis.

[Meta-Review · NeurIPS 2019]

The paper makes conceptual, algorithmic and theoretical contributions to optimization in multi armed bandits. It introduces a new problem motivated by settings where there is a testing phase with a cost structure proportional to the utility derived from playing each alternative, and gives a novel algorithm with a sample complexity bound for identifying a near-best arm. All the reviewers agree that the paper's contributions as above are significant. The only concern expressed was about the potentially narrow scope of the problem formulation, but the author feedback has helped in clarifying this aspect. It appears that the learning problem studied here, i.e., its cost structure specifically, could emerge in settings beyond e-commerce and advertising, and is likely to be of broader interest.